# RAS Dimers: The Novice Couple at the RAS-ERK Pathway Ball

**DOI:** 10.3390/genes12101556

**Published:** 2021-09-30

**Authors:** Ana Herrero, Piero Crespo

**Affiliations:** 1Instituto de Biomedicina y Biotecnología de Cantabria (IBBTEC), Consejo Superior de Investigaciones Científicas (CSIC)—Universidad de Cantabria, 39011 Santander, Spain; ana.herrero@csic.es; 2Centro de Investigación Biomédica en Red de Cáncer (CIBERONC), Instituto de Salud Carlos III, 28009 Madrid, Spain

**Keywords:** RAS, dimerization, RAS signaling, cancer

## Abstract

Signals conveyed through the RAS-ERK pathway constitute a pivotal regulatory element in cancer-related cellular processes. Recently, RAS dimerization has been proposed as a key step in the relay of RAS signals, critically contributing to RAF activation. RAS clustering at plasma membrane microdomains and endomembranes facilitates RAS dimerization in response to stimulation, promoting RAF dimerization and subsequent activation. Remarkably, inhibiting RAS dimerization forestalls tumorigenesis in cellular and animal models. Thus, the pharmacological disruption of RAS dimers has emerged as an additional target for cancer researchers in the quest for a means to curtail aberrant RAS activity.

## 1. Introduction

The RAS-ERK pathway encompasses a series of biochemical processes whereby signals are relayed from the plasma membrane to the interior of the cell in response to stimulation. Its main participants are classed as four types of proteins: the RAS family of small GTPases, and the kinases included in a three-tier module made up of RAF family serine/threonine kinases, MEK dual-specificity kinases, and extracellular signal-regulated ERK1/2 MAP kinases. This signaling route plays a central role in the regulation of key cellular processes, including proliferation, survival, differentiation, and motility. Thus, its unregulated activity, mostly as a consequence of genetic alterations, lies at the heart of the primeval events of cellular transformation and progression to full-blown neoplasia [1,2,3].

Initially, signaling through the RAS-ERK pathway was envisioned as a simple, linear sequence of events, in which, once RAS became activated in response to agonist stimulation, it fostered the phosphorylation and activation of RAF, which in turn phosphorylated MEK, which undertook ERK phosphorylation and activation. Decades of research have unveiled a rather more complex panorama, in which signal output is molded not only by the participating kinases but also by the intervention of ancillary regulatory proteins, such as scaffold proteins [4,5,6] and phosphatases [7,8,9,10]. In addition, the kinases that populate the cascade are able to oligomerize. Dimerization, in particular, adds one further level of complexity to the regulation of the pathway. Overall, the integration of the different regulatory mechanisms enables the fine-tuning of signal parameters such as frequency, amplitude, and intensity. This provides the system with both robustness and variability, making it possible for the same pathway to phosphorylate a wide spectrum of substrates and therefore regulate hundreds of cellular processes, yielding an ample variety of biological outcomes [11,12,13].

The RAS family is composed of four proteins, HRAS, NRAS, KRAS4A, and KRAS4B, which are encoded by three genes. Structurally, RAS proteins consist of identical N-terminal (residues 1–86) and C-terminal lobes (residues 87–171) with an 80% overall homology. The N-terminal lobe is considered to be the effector lobe, since it accommodates the P-loop (residues 10–14), switch I (residues 30–38), and switch II (residues 60–76) regions, which are responsible for effectors, GAPs, and GEFs binding. Both switches I and II are close to the GTP binding site, and the activation status of the protein implies a conformational change that determines the binding to effectors and regulatory proteins and also implies reorientation relative to the membrane [14,15,16]. The C-terminal lobe harbors the hypervariable region (HVR) that comprises amino acids 164–188/189 and contains the CAAX box, implied by RAS’s association with membranes (Figure 1). As a consequence of the postransductional modifications, the CAAX motif is the target of a series of modifications: the prenylation of cysteine to increase the hydrophobicity and its binding to the endoplasmic reticulum [17,18], the proteolysis of the −AAX residues [19,20,21], and the methylation of the carboxyl-prenylated cysteine [22,23]. To stably associate to membranes, some RAS proteins require a second anchoring signal—the addition of palmitate in the case of HRAS and NRAS [24,25], and the electrostatic interaction of the polybasic region of KRAS at its HVR with the phospholipids in the membrane [26].

## 2. Dimers, Dimers, Everywhere

Over the past few decades, evidence has accumulated demonstrating how oligomerization is a widespread phenomenon among the constituents of the RAS-ERK pathway. RAF family kinases are subject to exquisite regulation, the failure of which is at the heart of a wide catalog of pathologies, including cancer. It has been shown that RAS GTP loading induces both CRAF and BRAF dimerization [27], a key process in RAF activation. RAF kinase activity is boosted by its dimerization, which can be either homodimerization or heterodimerization among the three members of the family (ARAF, BRAF, and CRAF) [28].

Structural analyses have unveiled that RAF dimerizes through its catalytic domain, forming “side to side” dimers, and mutations in critical residues therein impede RAF-dependent MEK phosphorylation [29]. Likewise, peptides targeting the dimerization interface induced cell death in RAS-mutant tumor cells [30]. As such, RAF dimerization emerged as a promising target for therapeutic intervention—more so when it was found that such a process underlies the acquisition of resistance to BRAF kinase inhibitors in tumors harboring wild-type BRAF [31,32]. At this moment, new generations of compounds that target BRAF and CRAF homo- and heterodimerization are showing promising results in tumors refractory to previous types of RAF inhibitors [30,33,34].

MEK1 dimerization is necessary both for its activation by upstream RAF kinases and for ERK activation [35]. MEK1 and MEK2 form “head-to-head” homodimers, where the activation loop of one of the protomers aligns with the catalytic center of the other [36]. In addition, MEK1 and MEK2 can also heterodimerize, and in so doing contribute to the duration and intensity of MEK and ERK activity [37]. Interestingly, some types of tumors harbor highly prevalent MEK1 mutations, which map on the dimerization interface and enhance homodimerization [35]. Remarkably, enhanced dimerization confers resistance to the MEK inhibitors currently used in clinics, something that places MEK oligomerization in the limelight as a therapeutic target in cancer [35].

ERK1/2 dimers were the first to be described back in 1998. Both ERK1 and ERK2 can homodimerize, but, unlike the MEK and RAF families, ERK heterodimers are unstable [38]. The biochemical role played by ERK dimerization remains unclear. In this respect, it has been proposed to have an impact on ERK signal intensity [39]. Initially, dimerization was suggested to play a role in ERK nuclear shuttling [38], but it was later shown to be primarily involved in ERK cytoplasmic signaling, in association with scaffold proteins that serve as ERK dimerization platforms in order to phosphorylate cytoplasmic substrates [40]. As in the upstream tiers, the disruption of ERK dimerization prevents cellular transformation and tumor progression and dissemination, as demonstrated by DEL-22379, a small-molecule inhibitor that exhibits remarkable antineoplastic effects in tumor cells harboring RAS-ERK pathway oncogenes [41].

Overall, the picture emerging is that, even though its biochemical meaning still remains largely unexplained, the dimerization of the ERK cascade kinases appears to be an important process for carcinogenesis, and its inhibition can be exploited for therapeutic purposes. Data gathered over the past decade suggest that this may also be the case for RAS.

## 3. RAS Dimerization: A Biography

The first hint of RAS dimerization dates to 1988, when Santos and colleagues, using radiation inactivation assays, detected HRAS complexes of a molecular size compatible with clusters of two or more HRAS molecules [42]. Since then, efforts from many other groups have validated this initial observation while beginning to unravel the biochemical and physiological consequences of RAS dimerization.

While most aspects of RAS proteins have been the focus of exhaustive research for over 40 years, the study of RAS dimerization has not attracted real interest until recently. In 2000, Inouye et al. reopened the RAS dimerization case when they reported that RAF activation was dependent on the formation of RAS dimers, discovered by artificially forcing lipid-unattached RAS to form dimers using a bifunctional amine-reactive crosslinker [43]. Later, Prior et al., using immunogold electron microscopy of plasma-membrane “peel-offs”, described RAS multimers that formed clusters in different microdomains. Such RAS assemblies displayed an isoform-independent distribution [44], though it was not determined whether the RAS clustering was a consequence of RAS dimerization. Ten years elapsed before initial insights into the molecular structure of RAS dimers were available. The combination of Förster resonance energy transfer (FRET), attenuated total reflectance Fourier transform infrared (ATR-FTIR) spectroscopy, and biomolecular simulations—allowing for the measurement of molecule orientation and distance—yielded the first in vitro structure for NRAS, lipid-anchored and bound to artificial membrane bilayers to allow natural conformation during dimerization [45]. From this study, it was determined that NRAS oligomers were positioned perpendicular to the membrane, with such an orientation being stable only when RAS dimerizes. Similarly, the structure of membrane-bound HRAS associations was determined by applying fluorescence correlation spectroscopy (FCS) and time-resolved fluorescence anisotropy (TRFA) to measure molecular diffusion, and the photon counting histogram (PCH) and single-molecule tracking (SMT) to measure stoichiometry [46]. In combination, both techniques showed that HRAS oligomers were indeed dimers and not a higher-level complex, forming surface density-dependent clusters [46].

The KRAS dimer structure, both in a solution and in cells, was unraveled in two independent studies. On one side, it was shown that both GDP and GTP-bound KRAS G-domains formed dimers in the solution. FRET and nuclear magnetic resonance (NMR) spectroscopy provided insights into the stability of and distance between protomers, as well as the mapping of the dimerization interface, which turned out to be within the G-domain [47]. The KRAS-GDP/KRAS-GTP heterodimers exhibited a lower stability than the KRAS-GTP homodimers, the latter being one of the most abundant KRAS dimer species. In light of all these data, two different models of dimerization were proposed. The first one envisioned a β-sheet surface (involving β-2); however, this would impede binding to effector and regulator partners. The second model suggested an interaction between α-helices α3 and α4, which would allow binding to effectors and regulators [47]. This same group further analyzed the structures of KRAS and HRAS dimers and determined that the orientation of RAS proteins at the membrane governs the corresponding dimer’s structural conformation [48]. They proposed isoform-specific models, in which the membrane orientation affects the allosteric lobe (helices α3 and α4) for KRAS-GTP, as well as the interface formed by helices α4 and α5 in the case of HRAS-GTP [48].

A second team presented evidence for KRAS dimers using quantitative super-resolution photoactivated localization microscopy, which allows the localization and quantification of biomolecules in intact cells. The team also concluded that both KRAS-GTP and KRAS-GDP could form dimers. Constitutively GTP-bound KRASG12D dimers, but not monomers, were able to activate ERK signaling in vivo. It was estimated that a density of six RAS-GTP dimers per µm2 was enough to activate RAF [49]. By increasing the amount of KRAS molecules or by using an artificial dimerization system, KRAS clustering at the plasma membrane was enhanced, increasing signaling by inducing RAF activation. Conversely, lower KRAS expression levels or monomeric KRAS-GTP were not as efficient. This was the first in vivo demonstration of the relevance of RAS dimerization for the regulation of ERK signaling. From then on, conceptually, the road was paved for disrupting RAS dimers as a strategy to curtail signals through the RAS-ERK pathway [50,51]. Yet, for such an assignment, a precise characterization of the dimerization interface would be indispensable.

Even though most of the pertinent structural studies have suggested that all RAS isoforms form dimers [43,45,46,47,49], the detailed molecular structure of RAS dimers has been a matter of conflict. As recently reported by the Gerwert laboratory, the reported RAS dimer structures can be segregated into three main categories based on the regions involved in the interaction between monomers: (i) helix α4 /α5 dimers, (ii) helix α3/α4 dimers, and (iii) β-sheet interaction dimers [52]. In addition, a few studies [47,53] offered enough information on the atomic resolution to provide an accurate molecular description of the dimerization interface and dimer orientation. By using docking algorithms, FRET, electron paramagnetic resonance (EPR) spectroscopy, biomolecular simulations, and the experimental validation of predicted dimer structures, Rudack and colleagues presented a solid model for lipid-anchored and membrane-bound NRAS dimers. They attached a palmitoyl and a farnesyl group to cysteine 181 via a maleimide group and mixed the lipidated NRAS with 1-palmitoyl-2-oleoyl-sn-phosphatidylcholine (POPC) liposomes. In order to give more precise distance measurements between the two NRAS molecules, three different labeling positions were used: S106, T124, and the nucleotide binding site [52]. They demonstrated that the most likely NRAS dimer model corresponded to category I, where the dimer interface involved a helix α4 and helix α5 interaction (Figure 2). This study indicated that to form a dimer, a RAS monomer is slightly shifted with respect to its partner, forming a salt bridge between residues D154 and R161, and a second stabilizing, fluctuating interaction between residues E49 and H131. These residues are located in helix α4, helix α5, and the loop between β2 and β3 sheets. The dimerization interface was experimentally validated by mutagenesis to prove the role of the two key interacting amino acids involved in the formation of NRAS dimers. Switch I and switch II, the binding sites for GDP/GTP, RAF, GEFs, and GAPs, are on the opposite face to the dimerization interface and are therefore totally accessible for all functional and regulatory activities hitherto described for RAS. In this respect, it has been proposed that various dimerization modes may co-exist in nanoclusters [54], even though RAF activation would be mediated preferentially by the RAS dimers formed via the helical interface (α4 and α5 helices) interaction [52,53,55]. Importantly, these studies confirmed the existence of RAS dimers and provided reliable insights into the dimer molecular structure, while dismissing the formation of dimers as an artifact of increased protein levels or laser-induced interactions.

It is noticeable that the differences between the proposed structures for HRAS/NRAS and KRAS dimers could be explained by the differential orientation of HRAS/NRAS and KRAS at the plasma membrane, probably due to the differences in their HVR regions, as suggested by Jang et al. [56,57]. However, it must be noted that most of the RAS dimer crystal structures resolved hitherto emanate from crystallography and docking studies, where the HVR region and lipid anchors are cleaved [47,58,59]. Yet, it is believed that the HVR is required for RAS dimerization, since binding to membranes is an essential prerequisite [45,46,49,51]. Notwithstanding, as previously noted, the RAS G-domain has been shown to dimerize in a solution [47], and several residues have been experimentally validated as determinants in the interaction between RAS molecules [45,52,60,61]. The biological and biochemical significance of these residues in RAS dimerization does not necessarily imply that they are indeed part of the dimerization interface. These residues could be indirectly implicated in RAS dimerization—for example, by interacting with adaptor or scaffold proteins that indirectly facilitate the interaction between RAS protomers.

## 4. RAS Clustering and Dimerization

It has long been known that RAS needs to be anchored to the plasma membrane to be active. This seminal concept has grown in complexity and it is now clear that RAS is also functional at the endomembrane [62,63,64,65,66]. In addition, in any membrane system, RAS isoforms segregate differently at microdomains of distinct biochemical composition and physical–chemical properties [67,68,69]. A major advancement largely stemming from the work of John Hancock’s laboratory has been the discovery that, within these membrane microdomains, RAS forms associations, described as nanoclusters, consisting of groups of small numbers of RAS molecules [44,60]. These associations directly impact RAS lateral segregation and diffusion within the membrane and, concomitantly, its activity [70].

Due to their distinct HVRs, RAS isoforms undergo different post-translational modifications [17,19,20,23,25,26], which orchestrate their specific microlocalization and nanoclustering. Apparently, RAS nanoclusters are isoform-specific [44,60,69]. In addition to different cholesterol dependencies [44], the actin cytoskeleton also regulates RAS isoform preferences for membrane binding and clustering [71]. Upon actin cytoskeleton disruption, active KRAS (but not HRAS) nanoclusters are depleted [71]. Other components of the plasma membrane, such as caveolae, also influence the nanoclustering of RAS isoforms [70]. For instance, the depletion of caveolae facilitates KRAS clustering and compromises HRAS-GDP and -GTP lateral segregation [72,73]. RAS nanoclusters are also nucleotide-specific since GDP or GTP binding influences the orientation and lifespan of RAS at the membrane [70]. Being GTP-bound, H/KRAS, and presumably NRAS, are more stable—with a half-life of about one second—than GDP-bound RAS, which has a half-life of less than 0.1 seconds [71,74].

It is presently unknown whether RAS isoform differential clustering is a consequence of their sublocalization preference, or, contrarily, if their sublocalization dictates their differential nanoclustering. What seems to be clear is that the localization, from which the signal is initiated, shapes the nature of the signal [64,68,75], something that may contribute to the differential oncogenicity of RAS isoforms in specific cancer types.

It seems logical to think that RAS dimerization and clustering are related events. The question remains as to which one is the cause and which is the consequence. It is possible that RAS nanoclusters are the result of the accumulation and partitioning of RAS dimers in a specific microdomain. Contrarily, there is the possibility that the clustering of RAS molecules precedes and facilitates RAS dimerization by facilitating RAS interactions. In that respect, recent evidence leads towards the idea of dimerization-independent HRAS clustering, since HRAS dimerization inhibition does not alter HRAS’s binding to the plasma membrane [50]. Contrarily, the inhibition of KRAS dimerization results in a decrease in membrane-bound KRAS [50]. The difference between HRAS and KRAS clustering may be the consequence of their distinct post-transductional modifications [26].

Both RAS-GDP and RAS-GTP can bind membranes and form nanoclusters [44,71]. While there is evidence for GTP-dependent dimer stabilization, at least in a solution [47], low concentrations of KRAS-GDP dimers have been observed in intact cells [49]. Based on these observations, it could be that RAS-GDP nanoclusters represent a non-active, monomeric, and transient status prior to the formation of RAS-GTP nanoclusters composed of active, dimeric RAS. The possibility exists, however, that the observed RAS-GDP nanoclusters may be an artifact resulting from RAS overexpression and non-physiological accumulation. It has been suggested that RAS dimerization and clustering reduce its mobility and diffusion through the plasma membrane [76], partly as a consequence of dimerized/activated RAS recruiting other proteins such as effectors, regulators, and scaffold proteins, thereby increasing the molecular size of the activated complex, something that may even impact the mode in which RAS anchors to the membrane.

## 5. RAS Dimers’ Functional and Regulatory Interactions

As previously mentioned, recent findings suggest that RAS preferentially signals in a dimeric rather than monomeric form [49]. As reported, one monomer of KRAS-GTP would not be sufficient for RAF activation. The question remains whether this is extensible to all RAS isoforms, all RAS effectors, and all cellular contexts. It seems that the RAS dimer is the activation unit for RAF [43,49,50]. It is plausible that each KRAS-GTP molecule binds to one RAF monomer, and that RAS dimerization facilitates RAF proteins’ homo/heterodimerization. However, RAS dimerization itself may also be affected by downstream events. This is exemplified by the observation that RAF inhibitors, known to promote RAF dimerization, consequently evoke KRAS and NRAS but not HRAS nanoclustering by crosslinking constituent RAS monomers, thereby affecting the spatiotemporal dynamics of ERK signals [77].

Other effectors, such as RalGDS, have also been found to associate with RAS dimers [78]. This involves an interaction between the RAS-interacting domain (RID) in RalGDS and the switch I and II regions of HRAS [78]. Conversely, PI3K activation has been reported to be RAS-dimer independent. PI3K functions as a monomer and only needs one RAS molecule for its activation [79,80]. These differences in RAS dimer dependency for the activation of different effector routes may underlie the distinct effector usage that RAS displays depending on its subcellular localization [62,63,64,65,81].

RAS regulatory and other ancillary proteins may also participate to some extent in the RAS dimerization process. Galectin1 has been reported to bind HRAS and KRAS and to be essential for RAS binding to membranes and RAS-mediated transformation [82]. More importantly, Galectin1 is involved in RAS nanoclustering. In this respect, Gal-1 overexpression increases the size of RAS nanoclusters [83] and Gal-1 knockdown reduces the levels of RAS-GTP in such nanoclusters. The colocalization of Gal-1 and HRAS-GTP has been ascertained and shown to orchestrate HRAS signal duration via the stabilization of RAS nanoclusters [84]. However, whether Gal-1 HRAS-GTP participates in dimerization is a matter of controversy. In this respect, the Abankwa laboratory has proposed an alternative model in which Gal-1 interacts with the RBD (RAS-binding domain) of RAS effectors [85]. Based on this model, the stabilization of RAS assemblies would be posterior to RAS effector interactions and not necessary for RAS dimerization. 

## 6. RAS Dimers in Cancer

RAS plays a role as an oncoprotein during tumor initiation and progression, making it inevitable to enquire about the role played by RAS dimerization in oncogenesis. In this respect, recent findings on how RAS dimerization impacts tumorigenesis have opened new conceptual avenues towards new therapeutic strategies that may result in future treatments for RAS-driven cancers. 

A role for RAS dimerization in the aberrant activation of the RAS-ERK pathway, and the resulting tumorigenesis, has been revealed by the overexpression of DIRAS3 in cancer cells. DIRAS3, previously known as ARHI, is a small GTPase, initially reported as an antitumoral protein since its expression is downregulated in certain cancers, such as ovarian, pancreatic, lung, and breast cancers [86,87,88,89]. The mechanism by which DIRAS3 regulates tumorigenesis has been recently documented. DIRAS3 binds to RAS through its α5 helix, preventing RAS dimerization and consequent RAF activation. DIRAS3-mediated inhibition of RAS dimerization forestalls cellular proliferation and transformation [90]. The regulatory mechanism, whereby DIRAS3 couples and uncouples from RAS to allow its dimerization and signaling under physiological conditions, remains unveiled.

Efforts toward deciphering and characterizing the RAS dimerization interface have yielded the identification of the α4 and α5 helical domains as some of the main regions for the stabilization of RAS dimers [52,53,55]. Screening for monobodies aimed at RAS inhibition resulted in the discovery of NS1, a molecule that can bind HRAS and KRAS, but not NRAS, inhibiting RAS-induced ERK activation and cellular transformation [50]. NS1 binds to RAS on its dimerization interface via interactions with the α4 and α5 helices, specifically with residue R135. NS1 binding results in a decrease in RAS dimerization and membrane clustering, thereby impeding RAF activation. Furthermore, targeting the α4–α5 dimerization interface with an inducible NS1 monobody results in the inhibition of KRAS-driven tumor formation in vivo [91,92]. Intriguingly, mutations in residues R135, D154, and R161 within the RAS dimerization interface reduced NS1 binding but did not affect HRAS-mediated ERK signaling [50]. Recent studies, using in vitro phage display library screening, have revealed that DARPins (designed ankyrin repeat proteins) are able to bind both wild-type and mutant KRAS, but not HRAS or NRAS [93]. The interaction between DARPins and KRAS involves the α3 and α4 helices, impedes KRAS dimerization, and has a suppressive effect on mutant KRAS-driven cancer cells [93].

In vitro and in vivo experiments have also confirmed that the helical dimerization interface meddles in the formation of RAS dimers for RAF activation. Ambrogio et al. reported that interactions between residues D154 and R161 are essential for KRAS dimerization, and their mutation impairs RAS signaling and tumor growth [55]. The expression of a dimerization-deficient KRAS version, the D154Q mutant, revealed the prerequisite for KRAS dimerization for oncogenic KRAS-signaling in tumor progression. This work points also to RAS dimerization as the basis for wild-type KRAS’s inhibitory effect on mutant KRAS-driven cancers and also as the basis for MEK inhibitor sensitivity. On that matter, dimerization-deficient KRAS D154Q abolished the non-oncogenic KRAS-mediated growth inhibition and MEK inhibitor resistance. This work pointed to KRAS dimerization as a potential therapeutic target, since impairing KRAS dimerization leads to tumor growth reduction and to an increase in MEK inhibitor sensitivity [55]. 

In light of all these data, it becomes evident that the disruption of RAS dimerization as a monotherapy could serve as a valid strategy to curtail aberrant RAS signaling or, alternatively, as a complement to other RAS-ERK pathway-aimed therapies. For that, more research on the dimer structure, dimerization interface, interacting residues, and the regulatory mechanism governing RAS dimerization will be necessary (Figure 3).

## 7. Conclusions

Unquestionably, the RAS-ERK pathway is far from being a simple linear cascade of enzymatic reactions where every component has one defined task. It is now clear that every component of the pathway can function in higher-order associations, mainly as dimers, displaying differential activity depending on their oligomerization status. As such, dimerization could represent one more mechanism whereby a single route can be endowed with signal variability in order to provide a broad spectrum of outgoing responses to a large number of incoming stimuli.

In the past few years, the unprecedented development of sophisticated fluorescence-based microscopy techniques has proven essential for providing a wealth of data on RAS dimers’ structures and functions, which has paved the road toward evaluating their role in physiological signaling and carcinogenesis. While these fluorescence-based techniques will surely remain instrumental in unraveling RAS dimerization intimacies in the future, a major concern remains, in that studies involving genuine RAS dimers under physiological settings are still entirely missing. Thus, the question of to what extent ectopic RAS fluorescent chimeras resemble natural associations remains unanswered.

Dimerization-disrupting drugs are a current topic of interest. Small molecules interfering with ERK dimerization [41] and RAF dimer formation [33] have proven effective for curtailing oncogenic RAS-ERK signal flux. By inference, targeting the RAS dimerization interface offers a promising therapeutic venue, at least conceptually. Based on the current data, RAS dimerization inhibitors as a monotherapy could be effective for preventing mutant RAS aberrant signaling or could complement other therapies aimed at other downstream components such as MEK and RAF. In these and other biochemical struggles, RAS dimerization will surely provide food for thought in the near future.

## Figures and Tables

**Figure 1 genes-12-01556-f001:**
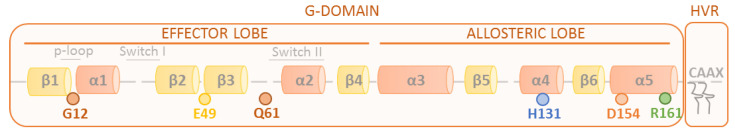
Schematic representation of the RAS structure. β-sheets (yellow) and α-helices (orange) are schematically shown in a linear representation of the RAS structure. Key RAS amino acids and regions involved in RAS oncogenicity (G12 and Q61) and dimerization (E49, H131, D154, and R161) are highlighted.

**Figure 2 genes-12-01556-f002:**
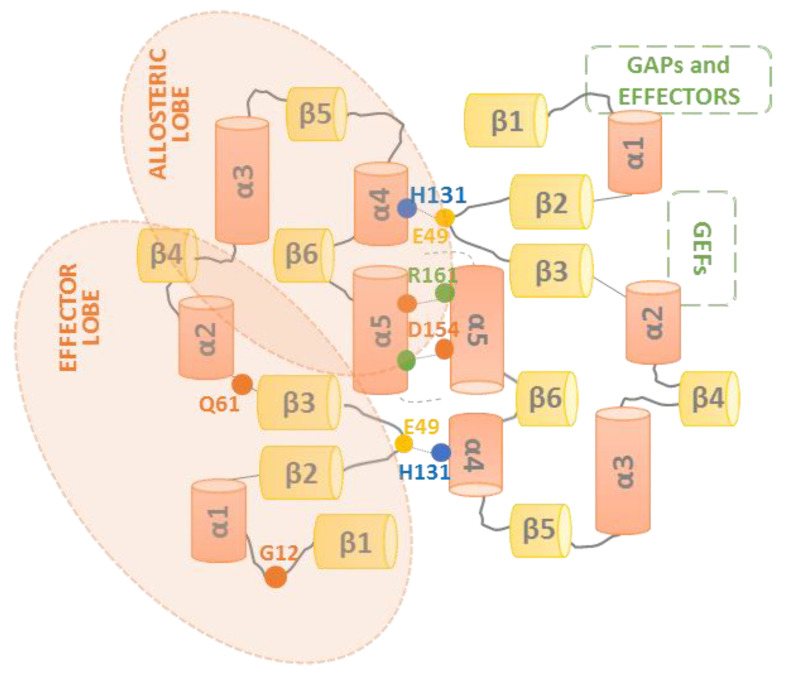
Representation of the RAS dimerization interface. β-sheets (yellow) and α-helices (orange) are schematically shown in a linear representation of the RAS structure. The dimerization interface of RAS dimers, comprising the α4 and α5 helices’ interaction, is represented, and the critical amino acids H131, E49, D154, and R161 (blue, yellow, green, and orange circles, respectively), involved in the interaction between two RAS protomers, are shown. The allosteric and effector lobes within the G-domain are highlighted, as well as the GEFs, GAPs, and effector binding sites (green squares).

**Figure 3 genes-12-01556-f003:**
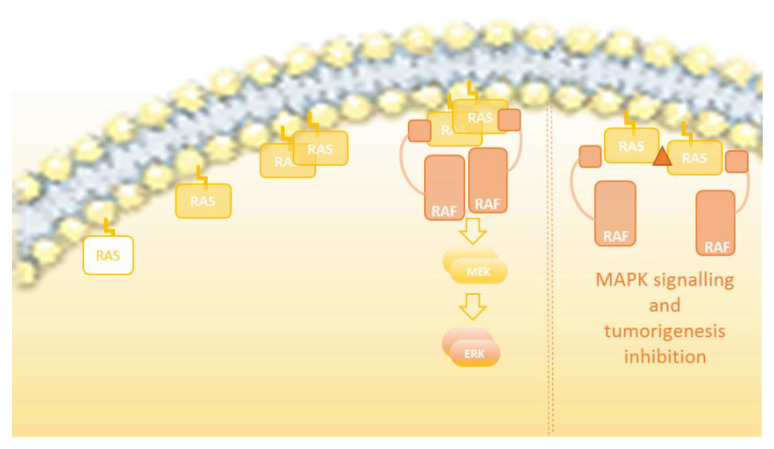
RAS-ERK pathway activation overview. The RAS-GDP monomer is bound to the membrane. Once the guanine nucleotide is changed to GTP and thereby activated, RAS-GTP dimerizes and is stabilized. The RAS dimers interact with RAF, facilitating its dimerization and consequent activation.

## Data Availability

Not applicable.

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
