# Peer review of "RAS Dimers: The Novice Couple at the RAS-ERK Pathway Ball"

_genes, 2021, doi:10.3390/genes12101556_

Round 1
Reviewer 1 Report
Herrero and Crespo present nice overview about the current knowledge of RAS dimerization. This is a less studied topic of the RAS field, but it is highly relevant as it can affect RAS activity and tumorigenesis. Overall, the review is well written and covers relevant literature. However, there are few comments that would need to be addressed.
It would help the reader to have better graphical description of the RAS structure to visualize better the different models. Figure 1 contains some of the information, but there is some missing information, like the localization of the nucleotide binding site or the RBD domain. Adding also some key aminoacids like G12, Q61 and other aminoacids that can help following the different dimerization models would also be helpful. The structure in figure 1 does not show that few aminoacids of the a5 helix are in the HVR region. This should be better represented as it is an important point considering that different dimerization domains have been proposed for the different RAS proteins.
Figure 1 is not cited in the text. I believe that after these changes are done it should be cited in the paragraph starting in line 117 after the first dimerization models are described.
As a minor comment, the amino acids indicated in the dimer structure in Figure 1 are very difficult to read. It would be helpful to increase the font size.
The paragraph starting in line 191 discusses the differences between the proposed structures for HRAS/NRAS and KRAS dimers. This is not discussed again until line 312 where the proposed model for KRAS seems to be the a4- a5 dimerization interfase. Is there a consensus proposed model for KRAS or this is still under debate? That should be better defined in the manuscript.
In the paragraph starting in line 289 the authors discuss the importance of Shoc2 and speculate its role in RAS dimerization. It has been described that Shoc2 forms a complex with MRAS and PP1 which plays a key role in RAF-ERK pathway activation by dephosphorylating a critical inhibitory site on RAF kinases and allowing the conformation changes required for RAF dimerization. I do not believe there is enough evidence to speculate the role of Shoc2 in RAS dimerization. Unless the authors can provide more recent literature that can support this hypothesis, this paragraph should be removed.
Minor comments
HVR is not defined in the text. It should be defined the first time that it is used.
The sentence “that the same pathway can regulate hundreds of cellular processes” in lines 40-41 is a bit misleading because it suggests that it is this fine control of pathway activation what results in the regulation of all these cellular processes. This can clearly play an important role. However, it is important to consider that ERK phosphorylates a big number of proteins, which explains why the RAS-ERK pathway alters so many processes.
In line 303, add that DIRAS3 was previously known as ARHI because all the references talk about ARHI.
In line 331, reference 88 does not fit. The sentence is still talking about the conclusions from reference 42, which is an article from 2018, whereas reference 88 is a review from 2005.
In Figure 2 the low contrast of white text in yellow squares makes it difficult to read.
Reference 75 is not properly cited. The journal and year are missing.
Line 15. Forestall is missing an s at the end.
Line 152. “a few” should be “few”
Author Response
Herrero and Crespo present nice overview about the current knowledge of RAS dimerization. This is a less studied topic of the RAS field, but it is highly relevant as it can affect RAS activity and tumorigenesis. Overall, the review is well written and covers relevant literature. However, there are few comments that would need to be addressed.
It would help the reader to have better graphical description of the RAS structure to visualize better the different models. Figure 1 contains some of the information, but there is some missing information, like the localization of the nucleotide binding site or the RBD domain. Adding also some key aminoacids like G12, Q61 and other aminoacids that can help following the different dimerization models would also be helpful. The structure in figure 1 does not show that few aminoacids of the a5 helix are in the HVR region. This should be better represented as it is an important point considering that different dimerization domains have been proposed for the different RAS proteins.
Figure 1 has been modified. It has been substituted by a more explicit Figure 1 and a new Figure 2.
Figure 1 is not cited in the text. I believe that after these changes are done it should be cited in the paragraph starting in line 117 after the first dimerization models are described.
Thank you for the observation. Figure 1 and 2 are correctly cited now.
As a minor comment, the amino acids indicated in the dimer structure in Figure 1 are very difficult to read. It would be helpful to increase the font size.
Indeed, there were very small. We have increased the font size in figures.
The paragraph starting in line 191 discusses the differences between the proposed structures for HRAS/NRAS and KRAS dimers. This is not discussed again until line 312 where the proposed model for KRAS seems to be the a4- a5 dimerization interfase. Is there a consensus proposed model for KRAS or this is still under debate? That should be better defined in the manuscript.
This is something under strong debate at this moment. We have included a comment in this respect.
In the paragraph starting in line 289 the authors discuss the importance of Shoc2 and speculate its role in RAS dimerization. It has been described that Shoc2 forms a complex with MRAS and PP1 which plays a key role in RAF-ERK pathway activation by dephosphorylating a critical inhibitory site on RAF kinases and allowing the conformation changes required for RAF dimerization. I do not believe there is enough evidence to speculate the role of Shoc2 in RAS dimerization. Unless the authors can provide more recent literature that can support this hypothesis, this paragraph should be removed.
Thank you for your comment. We have deleted the mentioned paragraph
Minor comments
HVR is not defined in the text. It should be defined the first time that it is used.
Corrected
The sentence “that the same pathway can regulate hundreds of cellular processes” in lines 40-41 is a bit misleading because it suggests that it is this fine control of pathway activation what results in the regulation of all these cellular processes. This can clearly play an important role. However, it is important to consider that ERK phosphorylates a big number of proteins, which explains why the RAS-ERK pathway alters so many processes.
Modified to: which make possible that the same pathway can phosphorylate a wide spectrum of substrates and therefore regulate hundreds of cellular processes, to yield an ample variety of biological outcomes
In line 303, add that DIRAS3 was previously known as ARHI because all the references talk about ARHI.
Included
In line 331, reference 88 does not fit. The sentence is still talking about the conclusions from reference 42, which is an article from 2018, whereas reference 88 is a review from 2005.
Corrected
In Figure 2 the low contrast of white text in yellow squares makes it difficult to read.
Modified
Reference 75 is not properly cited. The journal and year are missing.
Corrected
Line 15. Forestall is missing an s at the end.
Corrected
Line 152. “a few” should be “few”.
Corrected
We would like to thank the reviewer for all the comments and suggestions that have contributed to improve the quality of the manuscript.
Reviewer 2 Report
Outstanding review showing readers available data about the biological role of RAS dimerization in the context of the RAS-RAF-MEK-ERK cascade.
The review is punctual and updated and it clearly emerges authors’ deep knowledge about the issue analyzed in the manuscript.
Only some minor points should be addressed, according to my opinion, to make the manuscript even more brilliant:
- Please add a paragraph, maybe in the introduction or as independent section (up to the authors the decision), reporting structure of RAS and describing the different isoforms. Contextually, please introduce a dedicated and well-appealing figure about RAS structure.
- Please dedicate a separated paragraph to the tumors where RAS mutations have recognized pathogenic role in cancer development/progression. Among such tumors, thyroid cancers are involved, including the most common differentiated forms (differentiated thyroid cancer [papillary and follicular histotype]) and the less common medullary thyroid cancer. In such context, please cite the following papers:
Marotta V, Bifulco M, Vitale M. Significance of RAS Mutations in Thyroid Benign Nodules and Non-Medullary Thyroid Cancer. Cancers (Basel). 2021 Jul 27;13(15):3785. doi: 10.3390/cancers13153785. PMID: 34359686; PMCID: PMC8345070.
Marotta V, Sciammarella C, Colao AA, Faggiano A. Application of molecular biology of differentiated thyroid cancer for clinical prognostication. Endocr Relat Cancer. 2016 Aug 30. pii: ERC-16-0372. [Epub ahead of print] PubMed PMID: 27578827.
Moura MM, Cavaco BM, Leite V. RAS proto-oncogene in medullary thyroid carcinoma. Endocr Relat Cancer. 2015 Oct;22(5):R235-52. doi: 10.1530/ERC-15-0070. PMID: 26285815.
- Please add a further paragraph dedicated to the compounds blocking RAS dimerization and to available data from clinical trials analyzing their efficacy in tumors.
Author Response
Outstanding review showing readers available data about the biological role of RAS dimerization in the context of the RAS-RAF-MEK-ERK cascade.
The review is punctual and updated and it clearly emerges authors’ deep knowledge about the issue analyzed in the manuscript.
Only some minor points should be addressed, according to my opinion, to make the manuscript even more brilliant:
- Please add a paragraph, maybe in the introduction or as independent section (up to the authors the decision), reporting structure of RAS and describing the different isoforms.
Contextually, please introduce a dedicated and well-appealing figure about RAS structure.
Figure 1 has been modified. It has been substituted by a more explicit Figure 1 and a new Figure 2.
- Please dedicate a separated paragraph to the tumors where RAS mutations have recognized pathogenic role in cancer development/progression. Among such tumors, thyroid cancers are involved, including the most common differentiated forms (differentiated thyroid cancer [papillary and follicular histotype]) and the less common medullary thyroid cancer. In such context, please cite the following papers:
Marotta V, Bifulco M, Vitale M. Significance of RAS Mutations in Thyroid Benign Nodules and Non-Medullary Thyroid Cancer. Cancers (Basel). 2021 Jul 27;13(15):3785. doi: 10.3390/cancers13153785. PMID: 34359686; PMCID: PMC8345070.
Marotta V, Sciammarella C, Colao AA, Faggiano A. Application of molecular biology of differentiated thyroid cancer for clinical prognostication. Endocr Relat Cancer. 2016 Aug 30. pii: ERC-16-0372. [Epub ahead of print] PubMed PMID: 27578827.
Moura MM, Cavaco BM, Leite V. RAS proto-oncogene in medullary thyroid carcinoma. Endocr Relat Cancer. 2015 Oct;22(5):R235-52. doi: 10.1530/ERC-15-0070. PMID: 26285815.
Such paragraph would be irrelevant in the context of the present review. To our knowledge, there are no reports whatsoever making connections between mutant RAS pathogenicity in different tumours and its dimerization status. Specifically mentioning thyroid tumors would also be completely out of context.
- Please add a further paragraph dedicated to the compounds blocking RAS dimerization and to available data from clinical trials analyzing their efficacy in tumors.
There are only two studies, the best of our knowledge, reporting the use of compounds for RAS dimerization impairment. Work from Spencer-Smith et al., reported that NS1 monobody blocks and impedes HRAS and KRAS dimer formation, as mentioned in the text. In addition, Bery et al., have shown that DARPins (Designed Ankyrin Repeat Proteins) are able to bind α3 and α4 helix of KRAS, blocking its dimerization and activity. This work has been now included in the manuscript.
However, both compounds are large molecules not usable in clinical trials due to the limitations to cross the plasma membrane.
We would like to thank the reviewer for the comments and suggestions.